# A protocol for an economic evaluation of a polypill in patients with established or at high risk of cardiovascular disease in a UK NHS setting: RUPEE (NHS) study

Catriona Crossan,[1,2] Hakim-Moulay Dehbi,[3] Hilarie Williams,[4] Neil Poulter,[4] Anthony Rodgers,[5] Stephen Jan,[5] Simon Thom,[4] Joanne Lord[6]

[1]BresMed Ireland, Dublin 24, Ireland
[2]College of Health and Life Science, Brunel University London, London, UK
[3]CRUK& UCL Cancer Trials Centre, University College London, London, UK
[4]Peart-Rose Research Unit, International Centre for Circulatory Health NHLI, Imperial College London (Hammersmith Campus), London, UK
[5]The George Institute for Global Health, University of Sydney, Camperdown, Australia
[6]Southampton Health Technology Assessments Centre, University of Southampton, Southampton, UK

**Correspondence to**
Joanne Lord; jlord@soton.ac.uk

## ABSTRACT

**Introduction** The 'Use of a Multi-drug Pill in Reducing cardiovascular Events' (UMPIRE) trial was a randomised controlled clinical trial evaluating the impact of a polypill strategy on adherence to indicated medication in a population with established cardiovascular disease (CVD) of or at high risk thereof. The aim of Researching the UMPIRE Processes for Economic Evaluation in the National Health Service (RUPEE NHS) is to estimate the potential health economic impact of a polypill strategy for CVD prevention within the NHS using UMPIRE trial and other relevant data. This paper describes the design of a modelled economic evaluation of the impact of increased adherence to the polypill versus usual care among the UK UMPIRE participants.

**Methods and analysis** As recommended by the International Society for Pharmacoeconomics and Outcomes Research and the Society for Medical Decision Making modelling guidelines, a review of published CVD models was undertaken to identify the most appropriate modelling approach and structure. The review was carried out in the electronic databases, MEDLINE and EMBASE. 40 CVD models were identified from 57 studies, the majority of economic models were health state transition cohort models and individual-level simulation models. The findings were discussed with clinical experts to confirm the approach and structure. An individual simulation approach was identified as the most suitable method to capture the heterogeneity in the population at CVD risk. RUPEE-NHS will use UMPIRE trial data on adherence to estimate the long-term cost-effectiveness of the polypill strategy.

**Dissemination** The evaluation findings will be presented in open-access scientific and healthcare policy journals and at national and international conferences. We will also present findings to NHS policy makers and pharmaceutical companies.

## Strengths and limitations of this study

► This paper provides a clear outline of how a model for an economic evaluation is developed.
► Providing an outline of the model structure that includes details on the underlying epidemiology and data inputs will add transparency to the findings of the RUPEE-NHS study.
► Though the model has been designed to include all major adverse and beneficial effects of treatment, the model structure will not include every potential treatment effect.

in high-income countries.[3] Poor adherence is associated with greater deterioration in health status and increased healthcare costs,[4] and studies have shown that improved adherence to medication is associated with clinical benefits.[5] CVD preventive medication typically involves several drugs, and adherence is inversely proportional to the number of prescriptions. Furthermore, physician inertia and patient resistance present barriers to initiating or restarting full recommended therapy. A single pill that includes several indicated drugs (a 'polypill') may improve long-term adherence by addressing these issues. If the polypill is priced lower than the price of the pills bought separately, it will also make it more affordable.[6 7] The UMPIRE (Use of a Multi-drug Pill in Reducing cardiovascular Events) clinical trial was set up to evaluate the polypill in patients with or at high risk of CVD.

The UMPIRE trial randomised 2004 participants with established CVD (prior CVD event such as stroke or myocardial infarction (MI)) or at high risk of CVD (defined as a 5-year risk of >15%) based in India, England, Ireland and the Netherlands to either the polypill or usual

## INTRODUCTION

Adherence to recommended preventive medication regimes[1 2] in people at high risk of cardiovascular disease (CVD) is low, even

care. The primary outcome of the trial was adherence to indicated treatments (statin, aspirin and two blood-pressure-lowering drugs), measured as self-reported current use of antiplatelet, statin and ≥2 blood-pressure-lowering therapies for at least 4 days in the week preceding visits (baseline and end-of-trial visits). Other outcomes included systolic blood pressure (SBP) and low-density lipoprotein cholesterol (LDL-C). The trial found that the use of a polypill strategy resulted in greater adherence to treatment at 15 months and significant improvements in SBP and LDL-C. Detailed results and a description of the UMPIRE trial protocol are available.[8 9]

UMPIRE collected data on resource use and self-reported health-related quality of life using the EQ-5D. In order to estimate the long-term costs and health outcomes associated with the polypill strategy, an economic model is required. Due to differences in the patient population, care pathways and healthcare costs, separate analyses are needed for the four participating countries.

The analysis of the UMPIRE English trial data (Researching the UMPIRE Processes for Economic Evaluation in the National Health Service (RUPEE-NHS)) aims to estimate the cost-effectiveness of the polypill strategy compared with conventional multi-drug therapy for the prevention of CVD in English NHS patients with or at high risk of CVD. The RUPEE (NHS) study will use UMPIRE English trial data on adherence to the polypill and will develop an economic model to estimate cost-effectiveness.

The aim of this paper is to detail the modelling plan for the RUPEE (NHS) study.

## METHODS
### Model design process
An economic model has been described as a mathematical framework that represents reality at an adequate level of detail to inform clinical or policy decisions.[10] Guidelines on modelling produced by the International Society for Pharmacoeconomics and Outcomes Research (ISPOR) and the Society for Medical Decision Making (SMDM) joint taskforce recommend that it is best practice to carry out a conceptualisation process prior to programming the economic model. This process has two distinct components: specification of the study question and economic model.[11]

### Specification of the study question
The first component informs choices about how to structure the economic model and parameters.

The RUPEE (NHS) study aims to evaluate two different treatment strategies in a population with or at high risk of CVD. The population for the economic model is defined by the inclusion criteria of the UMPIRE trial.[9] The inclusion criteria are listed below:
► Aged ≥18 years and
► High CVD risk defined as either established atherothrombotic CVD (history of coronary heart disease

(CHD), ischaemic cerebrovascular disease or peripheral arterial disease (PAD)) or a 5-year risk of ≥15% calculated using the Framingham risk equation

The economic model will evaluate the polypill strategy compared with usual medication. In the UMPIRE trial, participants assigned to the polypill received one of two versions: version 1 contained 75 mg aspirin, 40 mg simvastatin, 10 mg lisinopril and 50 mg atenolol, and version 2 contained the same ingredients but substituted 12.5 mg hydrochlorothiazide for 50 mg atenolol. Participants assigned to usual care continued taking medications as prescribed by their general practitioner (GP).

The RUPEE (NHS) study will follow guidelines for modelling health technologies as recommended by the National Institute for Health and Care Excellence (NICE).[12] Therefore an NHS and Personal Social Services perspective will be adopted to measure health service resource use, and health-related quality of life will be measured by quality-adjusted life years (QALYs) obtained using the EQ-5D. As per the NICE guidelines, costs and QALYS will be discounted at a rate of 3.5% per year.[12] The time horizon reflected in the economic model will be lifetime to represent the chronic nature of CVD.

### Conceptualisation of the economic model
The second component of the conceptualisation process involves defining the economic model. There are two steps to this approach. The first step is to identify the appropriate modelling approach. The modelling approach defines the analytical framework of the economic model. Different types of analytical frameworks have been used to represent CVD including decision trees, state transition models, compartmental models, individual simulation models and hybrid models, which often combine elements from different frameworks.[13–17]

The second step determines the underlying structure of the analytical framework, which will represent the disease and care pathway. The modelling approach needs to reflect: (1) CVD and the care pathway for this population, (2) the beneficial and adverse effects of treatment (polypill or usual care) and (3) the impact of increased adherence to treatment on health outcomes.

The guidelines produced by ISPOR-SMDM on modelling recommend that existing models addressing related problems should be reviewed as this approach can help identify both the modelling approach and underlying structure.[11] To inform the RUPEE (NHS) economic model, we carried out a review of published models evaluating interventions for CVD.

### Review of published CVD economic models
The purpose of the literature review was to identify the appropriate analytical framework to represent the decision problem. The literature review also aimed to inform the underlying model structure: disease and care pathway.

### Search strategy

The search strategy was conducted using the NHS Economic Evaluation Database, the National Institute for Health Research Health Technology Assessment monograph series and the NICE guidelines website. The search terms used included 'cardiovascular disease', 'coronary heart disease', 'stroke', 'myocardial infarction', 'angina' and 'peripheral artery disease'. Studies were excluded from the review if they did not discuss the development or review of an economic model, if no disease states for CVD were included in the model and if the focus of the study was a diagnostic test or surgical intervention where the economic model used a time frame of <10 years. Studies were not excluded on the basis of intervention (drug treatment or lifestyle intervention) or on the basis of date published or language. We developed a data extraction form that included fields on model purpose, structure, health states and events, transparency and validation. We did not collate information about the findings of the model as the objective of the review was to identify alternative model frameworks and methods used to represent CVD.

An initial general literature search identified a 2006 systematic review of CHD policy models by Unal *et al*, which was updated in 2008 by Capewell *et al* and expanded to include stroke models.[17 18] The review by Capewell *et al* identified seven 'notable' CHD models (of which six had been identified in the previous review by Unal *et al*), nine stroke models and several models that were currently in development at the time of publication. We reviewed the notable models and models in development identified by Capewell *et al*. Citation searching of both systematic reviews was carried out to identify other models published since 2008.

### Review findings

Overall, 57 studies were identified that reported on 40 CVD models. Figure 1 presents the flowchart for the search strategy.

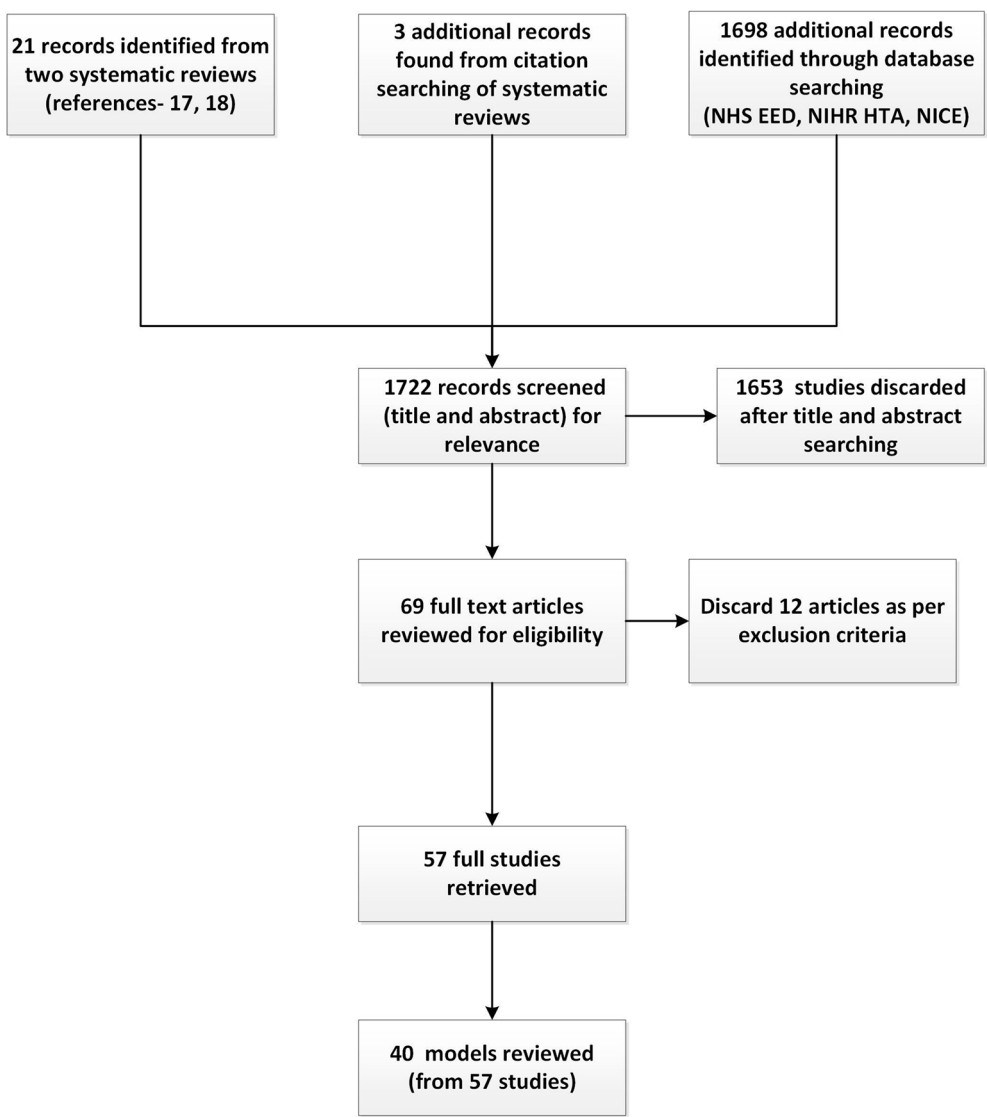

**Figure 1** Flow chart for search strategy for cardiovascular disease models. NHS EED, National Health Service Economic Evaluation Database; NICE, National Institute for Health and Care Excellence; NIHR HTA, National Institute for Health Research Health Technology Assessment.

The search found several studies that reported on the same model; for example, the IMPACT CHD model developed by Capewell *et al* was used in analyses of CVD in other populations.[19] In some cases, a model was adapted for different analyses, such as the Sheffield model, which was developed to evaluate statin therapy and was then adapted for use in the development of the NICE guidelines for lipid modification.[1 20] The Sheffield model was also partially used in a whole population modelling study by Barton *et al*.[13]

Further details of the review can be found in the online supplementary appendix. The appendix includes a list of the reviewed models (see table 1 and online supplementary appendix), an example of the data extraction form and an example of an illustration and details of one of the reviewed models (see figure 1 and online supplementary appendix). Schematic illustrations of several models were used in discussions with clinical experts about the different types of modelling approaches.

## Modelling approach

The search identified that the two most commonly used modelling approaches were health state transition cohort models and individual-level simulation models. Both approaches were critically assessed to determine their suitability to capture the disease and care pathway.

A cohort model can be defined as any model that estimates the outcomes for a group of patients, whereas with a patient-level simulation, outcomes are evaluated at the individual level. Therefore, one of the main differences between the two approaches is how they estimate costs and QALYs: cohort models estimate expected costs and QALYs for the modelled population as a whole, whereas individual-level simulation models estimate cost and QALYs for each individual and the average is taken across the sample.

With a health state transition cohort model, the population progresses through a set of mutually exclusive health states at regular intervals called cycles, determined by a predefined transition matrix. Health state transition cohort models are also commonly called Markov models. However, such models are only Markovian when they display the Markovian 'memoryless' property where the progression of the patient through the model is only dependent on the current state in which the patient resides and not on anything that happened before they entered that health state. It is also possible to model at the individual level using a state transition model by sampling probabilities for each individual patient to experience a particular transition in each model cycle.[21]

Both model approaches can use a discrete time approach: with this approach, the model cycle length will be defined in advance. The cohort or individual progress through health states or events that represent the disease pathway, and only one event may occur within each cycle length. Costs and QALYs are updated once per cycle. Alternatively, individual-level simulation models are often set up as discrete event simulations (DESs). With a DES approach, an event can occur at any time point; for example, an event could occur at 3 months, 1 year and 20 years. As an event occurs, costs and QALYs are recorded and updated for each individual.

A health state transition model was used to develop NICE guidance for lipid modification treatment.[1] The limitation of this approach is that it may be unable to capture the underlying heterogeneity in the population. Individual CVD risk can be estimated using CVD risk algorithms such as QRISK2 that use a range of patient characteristics such as age, sex, ethnicity, SBP and body mass index (BMI) to estimate a 10-year CVD risk.[22] To capture this complexity in a health state transition model would require the construction of a large number of subgroups to reflect different subsets of patient characteristics and the variation in CVD risk in the population. This could become impractical to model. It also has the disadvantage that accuracy could be lost by using representative values for subgroups. An individual simulation model structure may be more appropriate to model the level of detail required to estimate CVD outcomes reflective of those in the population.

The Markovian memoryless property means that data on individual patients are not retained as they progress through the model. Accounting for individual patient history in a Markov model would require multiplying the number of health states to an infeasible level where the model would become too complex.

To accurately identify the effectiveness of each treatment strategy in a population with or at high risk of CVD, an individual simulation model was deemed the most appropriate for the RUPEE (NHS) study to reflect the heterogeneity in the population, which impacts on the risk of a CVD event and subsequent costs and outcomes. The individual simulation model will use a discrete event approach to handle time.[21]

## Model structure

The findings of the review were discussed with clinical experts to confirm the health events and the methods used to model the progression of persons through the disease pathway.

### Model events (CVD, diabetes and adverse events)

The most commonly included types of CVD event in the reviewed models were CHD (angina and MI), cerebrovascular events (transient ischaemic attack (TIA) and stroke) and PAD. It was decided that the CVD events relevant for the current model would reflect those most commonly included in prior such models. PAD will not be included as a CVD event in the model as there is less likely to be a definable acute PAD event compared with other CVD events such as MI and stroke. We will assume that patients can experience more than one CVD event in their lifetime. The risk of CVD will also be assumed to change with age in the model.

**Table 1** Input paramaters

| Model inputs | Source |
| --- | --- |
| **1. Individual dataset** | |
| Population dataset | Initial patient characteristics (see figure 2) for cohort of patients drawn from a representative national sample: HSE dataset 2011. The dataset will include patients who meet the entry criteria for the UMPIRE trial |
| **2. Calculation of baseline risks** | |
| Risk calculators<br>Risk of heart failure | Risk of first CVD event and onset of type 2 diabetes estimated for individuals using QRISK2 and QDiabetes.[22 23]<br>*QRISK2: 10-year CVD risk (CVD outcomes defined as angina, MI, TIA and stroke)*<br>*QDiabetes: risk of acquiring type 2 diabetes over a 10-year period*<br>Risks for subsequent CVD events estimated for individuals using the REACH algorithm.[38]<br>*CVD outcomes defined as cardiovascular death (includes fatal stroke and MI), non-fatal MI, non-fatal stroke and cardiovascular hospitalisation (includes hospitalisation for unstable angina and TIA)*<br>Baseline risk per age using incidence rates in Cowie *et al*[49] |
| Relative incidence of CVD events (TIA, stroke, angina, MI) | OXVASC cohort study: 91 106 individuals presenting with an acute vascular event in Oxfordshire, UK, in 2002–2005[41] |
| **3. Adherence to medication** | |
| Probability of adherence to treatment with usual care | Estimates from HSE 2011 dataset on adherence to relevant drugs (statins, antihypertensives, aspirin) |
| Relative risk of adherence: polypill versus usual care | Estimate the probability of adherence to≥ 2 antihypertensives, statin or antiplatelet for at least 4 days in the preceding week for polypill group versus usual care by applying a binomial regression to the UMPIRE dataset |
| **4. Treatment effects of medication (antihypertensives, statin, antiplatelet)** | |
| Relative risk of CVD with treatment versus no treatment | For base case analysis, conventional meta-analysis of ITT RCT data will be used from:<br>▶ Cholesterol Treatment Trialists' Collaboration[50]<br>▶ Blood Pressure Lowering Treatment Trialists' Collaboration[28]<br>▶ Antithrombotic Trialists' Collaboration<br>▶ Law, Morris and Wald[51]<br>Sensitivity analysis: test impact of adjusting for adherence within trials |
| **5. Other treatment outcomes (beneficial events and adverse events) and mortality rates** | |
| *Adverse events* | |
| Incident type 2 diabetes | Relative risk of diabetes from statins/antihypertensives from meta-analyses of RCTs[24–27] |
| GI bleeding | Relative risk of bleeding resulting from aspirin using estimates from meta-analyses of RCTs[28] |
| Cough | Placebo-adjusted relative risk of cough resulting from ACE inhibitors using estimate from meta-analyses of RCTs |
| Reduction in heart failure | Relative risk reduction in heart failure from antihypertensives[33] |
| *Mortality* | |
| Stroke case fatality (60 days) | |
| Age<75<br>Age>75+ | Estimate proportion of strokes that are fatal (with risks increasing with age). Estimate using the BHF Compendium of Health Statistics 2012, which has data from a record linkage study for England 2010 |
| MI case fatality (30 days) | |
| Age 30–54 | Proportion of MIs that are fatal. Estimate using Oxford Record Linkage pill study.[52] National-population-based study, including all individuals admitted to hospital or who died suddenly from acute MI in 2010. Age was strongest predictive factor for 30-day case fatality |
| Age 55–64 | |
| Age 65–74 | |
| Age 75–84 | |
| Age 85+ | |
| Death from other causes | Estimated from national life tables (Office for National Statistics, England)[42] |
| **6. Costs (medication, monitoring costs, health events)** | |
| *Drug costs (£ per year)* | |
| Statins | National Health Service (NHS) Electronic Drug Tariff[44] |
| AHT drugs | |

| Table 1 Continued | |
|---|---|
| Model inputs | Source |
| **Aspirin** | |
| Polypill | Assumed to be aggregate cost of each drug in the combined pill |
| **Yearly monitoring costs while on medication** | |
| Primary care nurse (£ per hour) GP cost (£ per hour) Lipid test (£) Liver transaminase test Blood tests Costs of health states and adverse events | Use NICE Quality Outcomes Framework to identify recommended management while on treatment (statins, antihypertensives, antiplatelet). A cost for stopping medication will also be applied (eg, two GP visits, tests as recommended in NICE clinical guideline 181)[1] Costs sourced from Personal Social Services Research Unit Costs and NICE clinical guideline 181 |
| Stroke TIA | Luengo-Fernandez et al[53] |
| MI Angina PAD Diabetes | NICE lipids guideline 181[1] |
| **GI bleeding** | |
| Cough (from ACE inhibitor use) | NICE Hypertension guidelines 127[43] |
| **7. Health-related quality of life** | |
| Stroke TIA MI Angina MI PAD GI Bleeding Diabetes Cough | Derived from HSE dataset |

AHT, antihypertensive; CVD, cardiovascular disease; GP, general practitioner; HSE, Health Survey for England; ITT, Intention to treat; MI, myocardial infarction; NICE, National Institute for Health and Care Excellence; RCT, randomised controlled trial; TIA, transient ischaemic attack; UMPIRE, Use of a Multidrug Pill in Reducing cardiovascular Events.

Diabetes is a risk factor for CVD with a substantial cost and impact on health-related quality of life; therefore, diabetes will be included as a comorbidity in the model. The risk of new onset diabetes will be estimated using the QDiabetes risk algorithm.[23]

Adverse effects from treatment will include an increase in the risk of new onset diabetes resulting from treatment with statins and antihypertensive drugs.[24–27] The risk of a persistent cough resulting from treatment with ACE inhibitors will be included as an event. The probability of a cough resulting from treatment will be sourced from meta-analyses of randomised controlled trial (RCT) data for ACE inhibitors. As aspirin use is associated with an increased risk of gastrointestinal bleeding,[28 29] an increased risk of gastrointestinal bleeding from treatment with aspirin will be included.

Renal impairment will not be included in the model as an adverse effect of ACE inhibitors. While ACE inhibitors may cause an acute rise in serum creatinine in a few patients with renal artery stenosis and more generally cause a slight short-term increase in creatinine levels, the effects are complex and there may be a net improvement in renal function overall in a treated population. The rate of falls and fractures will be estimated not to alter, given the evidence from randomised trials of blood-pressure-lowering agents, although this is an area of debate with regard to patients with higher levels of frailty.[30 31]

Other adverse effects from statin treatment such as liver dysfunction and myopathy will not be included in the model as severe cases are rare.[1 32]

Treatment with antihypertensives is associated with a reduction in heart failure; therefore, this will be included as an outcome in the model.[33] Other outcomes of treatment are likely but will not be included, for example, a reduction in cancer with aspirin use of more than 5 years.[34]

### Progression of individuals through model

The progression of persons through the disease pathway differs depending on the modelling approach: health state transition models such as the Markov model developed for NICE guidelines on lipid modification use a predefined transition matrix to determine progression through the CVD health states.[35] Alternatively, simulation models can use risk algorithms to estimate the probability of CVD events or new onset diabetes. The NICE guidelines for lipid modification recommend the use of QRISK2, which is a risk algorithm derived to estimate primary CVD risk in UK populations.[1 22] The QRISK2 risk algorithm predicts the risk of a 10-year CHD event

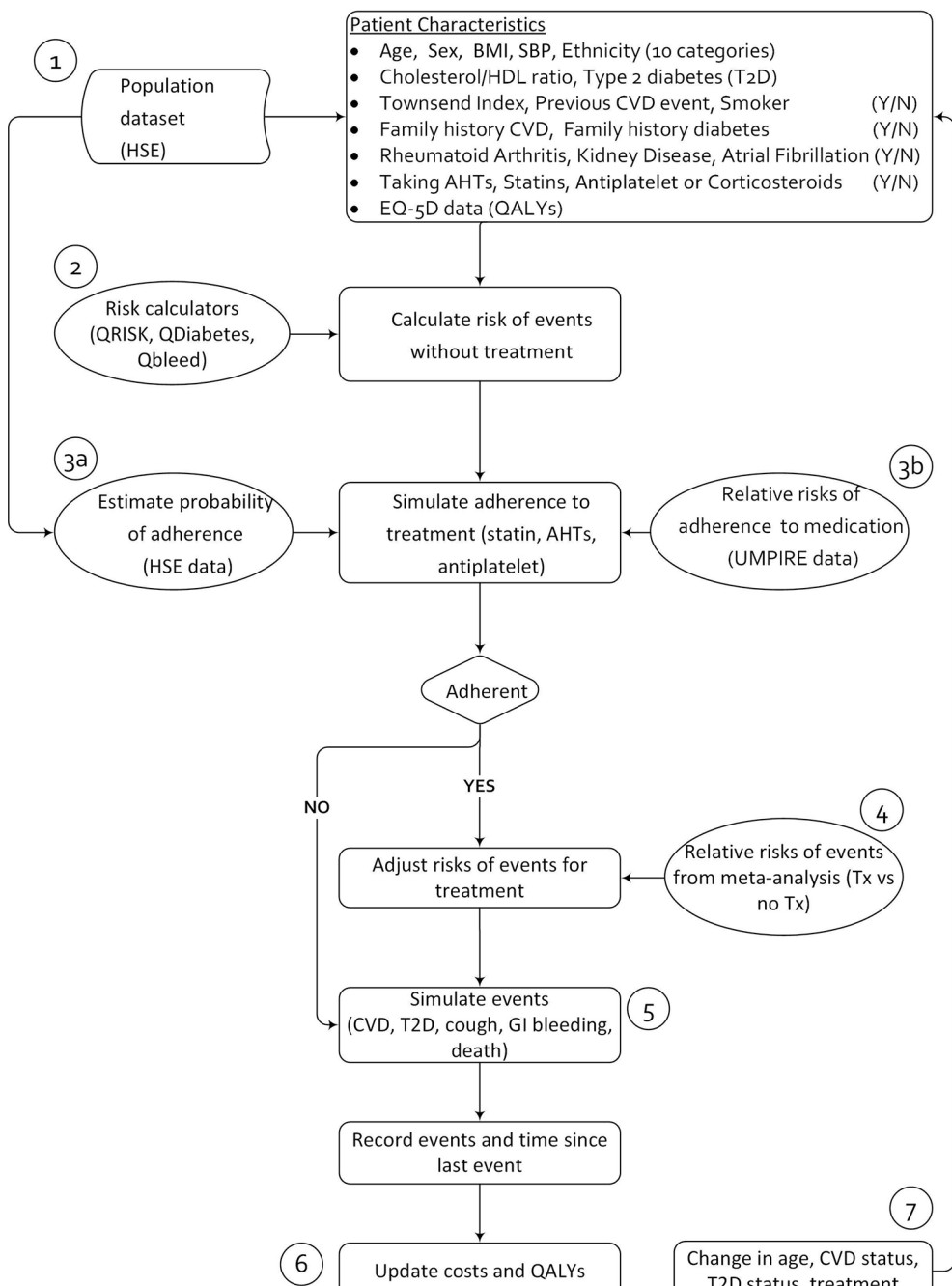

**Figure 2** Flowchart of RUPEE (NHS) model structure. AHT, antihypertensive; BMI, body mass index; CVD, cardiovascular disease; GI, gastrointestinal; HDL, high-density lipoprotein; HSE, Health Survey for England; QALY, quality-adjusted life year; RUPEE-NHS, Researching the UMPIRE Processes for Economic Evaluation in the National Health Service; SBP, systolic blood pressure; tx, treatement.

(angina, MI) or a cerebrovascular event (TIA, stroke). It does not include the risk of PAD. An alternative CVD risk algorithm is the Framingham equation;[36] however, a validation study comparing QRISK2 and Framingham found that QRISK2 is better calibrated to a UK population.[37] The RUPEE (NHS) model will therefore use the QRISK2 risk algorithm.

## RUPEE (NHS) ECONOMIC MODEL

Figure 2 depicts the flowchart of the RUPEE (NHS) model structure. The oval shapes represent data inputs

to the model, whereas the rectangular shapes represent processes.

Model description

In the RUPEE (NHS) model, costs and QALYs are recorded for each individual and an average cost and QALY for the simulated population are estimated. The RUPEE (NHS) model will be run twice, once to simulate costs and QALYs under usual care and once to simulate costs and QALYs under the polypill scenario (polypill scenario will include polypill version 1 and version 2). Individuals representing the UMPIRE trial inclusion

criteria will enter the model (label 1 in figure 2), and their baseline risk of a CVD event and new onset diabetes will be estimated using the QRISK2 CVD risk algorithm and QDiabetes algorithm (label 2 in figure 2), respectively. For each individual, whether or not they are adherent to medication will be simulated using Monte Carlo simulation based on the probability of adherence in usual care (label 3a in figure 2). If the individual is simulated to be adherent to medication, their risk of a CVD event will be modified by a treatment effect (label 4 in figure 2). In the polypill scenario of the model, the probability of adherence will be further modified by the relative risk of adherence to medication. The relative risk of adherence to medication will be sourced from the UMPIRE trial data (label 3b in figure 2). Individuals may experience a CVD event or onset of diabetes based on their estimated CVD and diabetes risk, which will be estimated using the QRISK2 and QDiabetes algorithms. Individuals may also experience an adverse reaction to medication (if adherent) including gastrointestinal bleeding, early onset of diabetes and a persistent cough. Costs and QALYs will be recorded for each event (including adverse events). Individuals can experience more than one event (model run for lifetime horizon), and patient characteristics such as age and history of previous events, such as a stroke or new onset diabetes, are updated during the model run, with an ensuing reflective increase in the risk of an event.

### Input parameters

Each point in the flowchart is labelled, and a description of the process or data requirement label is described below. Table 1 provides further details on data input parameters for the RUPEE-NHS model and potential sources of data.

### Population dataset

We will use the 2011 Health Survey for England (HSE) as a population dataset for the economic model. The HSE is a cross-sectional survey that contains anonymised information on a representative sample of the population. The 2011 HSE dataset collected information on CVD, including individual CVD events and medication history. The dataset also contains information on demographic and socio-economic characteristics and health-related data such as BMI, SBP and LDL-C and history of CVD events. These data are required in order to estimate individual baseline risks of CVD and diabetes in the model.

### Calculation of baseline risks of events without treatment

Baseline risks for CVD for each sampled individual will be calculated using published risk algorithms. As per recent NICE guidance for lipid modification, we will use the recommended algorithm for CVD risk, QRISK2.[1 22] The algorithm was derived using QRESEARCH, a large database derived from the pseudonymised health records of over 13 million patients registered with a GP in the UK. If an individual has established CVD (previously experienced a CVD event), we will estimate a secondary CVD risk using the REACH algorithm.[38] A baseline risk for the

onset of diabetes will be estimated using the QDiabetes algorithm.[23]

### Simulating adherence to treatment under usual care

The RUPEE study will evaluate the effect of adherence to medication on long-term costs and health outcomes measured using QALYs. The average rates of adherence in clinical trials can be higher than in actual practice[4] as seen in the UMPIRE clinical trial population, which had an atypically high baseline adherence rate. Instead, adherence rates to medication (antihypertensives, statins and aspirin) under a usual care setting will be sourced from the 2011 HSE dataset. Participants in the 2011 HSE self-reported all the prescribed medications they had taken in the last 7 days. This was coded in the HSE dataset using the British National Formulary classifications codes. Using these data, we are able to identify the medication patients were prescribed and identify whether or not they were taking the prescribed medication in the last week. This will reflect adherence to medication in a usual care population. The data will be used to estimate the probability of each person being adherent or not to medication. Individual characteristics will be used as predictors of adherence; the characteristics will be chosen by referring to studies that have assessed predictors of adherence in persons taking treatment for CVD.[39 40] A generalised linear mixed regression model will be used to estimate the probability of adherence to medication for each individual. The probability of persistence with medication will not be assumed to be constant, and the model will include a probability of ceasing medication over time. The probability of medication cessation will be sourced from published literature on adherence.

### Estimate relative risks of adherence to medication

We will estimate the relative risks of adherence to medication, using a generalised linear mixed regression model, which will be applied to the UMPIRE trial dataset (UK dataset). In the polypill scenario in the model, the probability of being adherent to medication will be further modified by the relative risks.

### Adjust risk of events for treatment

We will source data on the treatment effects of statins, antihypertensives and aspirin from meta-analyses of intention-to-treat (ITT) RCTs. ITT analyses account for non-adherence in their findings and therefore underestimate the impact of treatment on event risk. To overcome this, we will carry out sensitivity analyses to test the impact of adjusting for adherence within the trial. The risk of a CVD event will be adjusted by the relative risk of treatment with statins, antihypertensives and aspirin, based on the medication(s) the person is taking and whether or not they are adherent to medication.

### Simulation of events

Individuals in the model can experience a CVD event at a rate governed by their calculated baseline risk (estimated by the QRISK2 or REACH algorithms) and

adjusted for treatment effects if they have been simulated as adherent to treatment. CVD events will be categorised as a TIA, stroke, MI or angina. The relative incidence of each CVD event will be determined using published incidence data.[41] Similarly, the risk of new onset diabetes will be calculated using the QDiabetes algorithm. We will simulate the incidence of adverse events as a result of treatment: new onset diabetes and gastrointestinal bleeds. Data on the probability of an adverse event will be sourced from meta-analyses of RCTs for the relevant drugs. Mortality risk will be modelled as mortality from stroke and MI and other cause of mortality. Data on other cause mortality will be estimated using national life tables for England and Wales.[42]

### Assign cost and quality of life values

Costs and QALYs associated with each individual's simulated lifetime profile of CVD and related care will be estimated. Costs and QALYs will accrue for each person to reflect events, such as a stroke or new onset diabetes. Costs and utility values for health events will be sourced from published studies including the NICE guidelines for lipid modification and hypertension.[1 35 43] Costs of medication will be sourced from the NHS National Drug Tariff.[44]

### Change in age, treatment, CVD status and type 2 diabetes status

The simulation model will run for each individual for lifetime duration (death or maximum age of 100 years), and patient characteristics will be updated after each event or every 10 years (depending on which occurs first). A 10-year update is used as the QRISK2 algorithm returns a 10-year CVD risk.

### Analysis

The simulation model will run for a sufficient number of iterations to provide stable results. Uncertainty in the model parameters will be examined using a probabilistic sensitivity analysis (PSA), which will reflect uncertainty over the values of the model inputs. Non-parametric bootstrapping of HSE data will be carried out to examine the uncertainty related to the sampling. For each PSA iteration, one non-parametric bootstrap sample will be drawn from the HSE dataset (by random sample with replacement of individuals in the dataset). An incremental analysis will be conducted and incremental cost-effectiveness ratios and net benefit statistics will be estimated. We will also carry out a number of sensitivity analyses to test the impact of varying uncertain parameters in the model. This will include an analysis testing the impact of varying the polypill cost.

### Validation

The model will be internally and externally validated. A checklist produced by the RUPEE steering group based on current published guidelines for checking models will be used, to ensure that the programmed model behaves as expected according to the theoretical model.[21 45] The checklist includes tips for model developers, for example,

on the use of sensitivity analyses to test that the model is operating correctly, and reprogramming complicated sections of code in another language. The model will also be reviewed and tested by an experienced modeller. The model results will be compared with real-world observations or the results of other models.

### Dissemination of results

The findings of the economic evaluation will be presented to scientific and healthcare policy audiences in open-access journals and at national and international conferences. We will also present findings to NHS policy makers and pharmaceutical companies.

## DISCUSSION

Medication adherence is important for disease management, and benefits of increased adherence to preventative medication for CVD include improved clinical outcomes.[5] The UMPIRE clinical trial was conducted to evaluate the effect of a polypill strategy compared with usual care on adherence. It showed that the polypill strategy significantly augmented adherence, and this was reflected by improvements in SBP and LDL-C.[8] Whether or not this impact remains in the long term cannot be determined from the trial data alone. The RUPEE (NHS) study is being conducted to evaluate the long-term impact of a polypill strategy; in particular, the analysis will evaluate the long-term impact of increased adherence on outcomes. An economic model is being developed to estimate the long-term costs and QALYs associated with implementing a polypill strategy in the NHS compared with usual care. This analysis will represent the first comprehensive cost effectiveness analysis using directly applicable clinical trial data.

This paper outlines the process behind the design of the economic model. We carried out a review of published CVD models to identify a modelling approach that would suit the healthcare decision: use of a polypill versus usual care in a population with or at high risk of CVD. We identified an individual simulation model as the most appropriate approach as it allows the heterogeneity in the population to be adequately reflected. The model will use validated disease risk algorithms to estimate the probability of an individual experiencing a CVD event or the onset of diabetes. Individuals can also experience an increased risk of an adverse event (diabetes, cough and gastrointestinal bleeding) from treatment. The risk of a CVD event will be reduced if the individual is adherent to treatment. We will simulate adherence to treatment using data from the HSE 2011 dataset. The probability of adherence in the polypill scenario will be further modified by the relative risks of adherence to medication, which will be sourced from the UMPIRE trial data for the English population. Costs and QALYs will be estimated for each individual and aggregated across the sample population (based on the HSE 2011 dataset).

The RUPEE (NHS) model will have a number of advantages over existing models constructed to evaluate a CVD polypill.[46–48] One advantage is the use of an individual simulation model, which will allow us to capture the heterogeneity in the variation in CVD risk in the UK population unlike other models that use Markov-type transition state models. Another is that we will extrapolate data on adherence to medication from a nationally representative population dataset (HSE), which will allow us to simulate adherence per individual rather than assuming a constant adherence across our population. We will also allow for adverse events from treatment and treatment cessation, therefore more accurately reflecting clinical practice.

It would be preferable to use per protocol treatment effectiveness data in our analysis as ITT data already account for adherence (people switching and ceasing medication during the trial period). However, per protocol data are difficult to obtain for all drugs; therefore, we will use the ITT treatment effect data and carry out sensitivity analyses to test the impact.

The introduction of a CVD preventive polypill strategy will simplify pill taking for patients, potentially leading to greater adherence and better health outcomes. This analysis will provide information on the cost-effectiveness of the polypill in an NHS setting.

**Acknowledgements** All research was carried out while CC was based at Brunel University London.

**Contributors** CC carried out the literature review and drafted the manuscript. CC, H-MD, JL, ST, HW, NP, AR and SJ contributed to the development of the protocol. JL provided input on the heath economics model. H-MD provided statistical advice, and ST, HW, NP, AR and SJ contributed clinical advice. AR and SJ peer reviewed draft manuscripts and contributed to the final version of the protocol. All authors approved the final version of the manuscript submitted for publication.

**Funding** This paper presents independent research funded by the National Institute for Health Research (NIHR) under its Research for Patient Benefit Programme (Grant Reference Number PB-PG-1112-29080).

**Disclaimer** The views expressed are those of the author(s) and not necessarily those of NHS, NIHR or the Department of Health.

**Competing interests** ST, NP and HW acknowledge support from the National Institute for Health Research Biomedical Research Centre at Imperial College Healthcare NHS Trust and Imperial College London. SJ and AR are employed by The George Institute for Global Health who own the IP for the polypill in Australia.

**Provenance and peer review** Not commissioned; externally peer reviewed.

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
