## [Reviewer comments · BMJ Open]

ARTICLE DETAILS

TITLE (PROVISIONAL)	A Protocol for a modelled economic evaluation to evaluate a polypill in patients with established or at high risk of cardiovascular disease: Researching the UMPIRE Processes for Economic Evaluation in the UK National Health Service - RUPEE (NHS)
AUTHORS	Crossan, Catriona Dehbi, Hakim-Moulay Thom, Simon Poulter, Neil Williams, Hilarie Rodgers, Anthony Jan, Stephen Lord, Joanne

VERSION 1 - REVIEW

REVIEWER	Kathleen Bennett
REVIEW RETURNED	12-Jul-2016

GENERAL COMMENTS	This is a well written protocol for an modelled economic evaluation to evaluate the polypill in patients with established or high risk CVD using information from an existing trial UMPIRE applied to the UK NHS. The authors have followed ISPOR modelling guidelines in their economic evaluation which is to be commended. However, the study is only applicable to the UK NHS when the original UMPIRE trial was conducted in India, Ireland and the Netherlands in addition to the UK. It would have been useful to see how a similar process may be applied to these other countries to make this more generalizable. Of course, different data sources would be used for some of the inputs but there would be some similar data sources. Methods The inclusion criteria refer to those at high risk of CVD using the Framingham risk equation, however, later in the methods the QRisk2 is referred to as the preferred (better calibrated to the UK population, page 9) and it would be useful to reconcile why the different scores are used here. It was not clear how adherence was measured within the trial or as the outcome in this model? Perhaps this needs to be explained further. The search strategy for models included a number of databases but was this sufficiently wide enough? Further details on the review would have been helpful, even as a supplementary appendix. Also very little on the findings on the review were presented – instead the results suggest that the individual-level model simulation is the preferred method here. On page 8 adverse effects are referred to, but what about the different severity or consequences of these effects and combined
--

	effects of more than one adverse effect (or combined effect of multiple therapies on each AE)? Two different dosing strategies are mentioned for the polypill (version 1 and 2)– but the model described refers to one scenario of the polypill – how are the findings from the two versions to be combined? One version contains a diuretic instead of atenolol. These might expect to have different adverse effect profiles etc. How large is the HSE cross-sectional survey – further information on the representativeness of this for the model would be helpful. Was the adherence data self-reported from this ? There are reviews examining the predictors of adherence in CVD that could be referred to. Section 4. ‘the risk of a CVD event’ will be adjusted by the relative risk of treatment, based on the treatment they are taking’. This was not clear – what treatment? Within the polypill? Other treatments patients may received? Table 1 – For adverse events will the RR will be compared to ‘no’ treatment option in each case? MI case fatality – based on data from one region in UK – is this sufficiently representative or has been used previously? Costs of health states – some information given but not for all. Figure 1 – need to expand legend to Flowchart of literature review on CVD models perhaps. Also, as mentioned above no information provided on what the review found. Figure 2 has no legend and this needs to be included. Please provide full text for abbreviations used as footnotes etc.
--	--

REVIEWER	Pelham Barton
REVIEW RETURNED	13-Jul-2016

GENERAL COMMENTS	This manuscript describes the plans for a model-based economic evaluation of a polypill. As such, it is appropriate material for BMJ Open. The manuscript appears to be very well written, and I have only a few minor points to raise. Page 4 lines 8-13: The guidelines mentioned were produced not by ISPOR acting as a body, but by a joint taskforce from ISPOR and the Society for Medical Decision Making. The description here should make that point clearly. Later (and in the abstract), it would be appropriate to refer to "the ISPOR-SMDM guidelines". Page 4 line 46 should presumably be reference 12 not 11. Page 7 lines 45-51: The reference to tunnel states is not really appropriate. Tunnel states deal with the specific issue of allowing the transition probabilities to vary according to time in a state. It would be better to make the general point that accounting for patient history requires multiplying the number of states to an infeasible level. Page 8 line 15: PAD needs definition. Page 11 line 9: the abbreviation AHT appears to be used only in Table 1 but never in the text. Accordingly, it should only be defined in the footnotes to the table, not here. Table 1 footnote correctly repeats the definitions of some
---

	abbreviations given in the text, but not all of them. All abbreviations used in the table should be defined in its footnote.
--	--

VERSION 1 – AUTHOR RESPONSE

Reviewer: 1

This is a well written protocol for an modelled economic evaluation to evaluate the polypill in patients with established or high risk CVD using information from an existing trial UMPIRE applied to the UK NHS.

The authors have followed ISPOR modelling guidelines in their economic evaluation which is to be commended. However, the study is only applicable to the UK NHS when the original UMPIRE trial was conducted in India, Ireland and the Netherlands in addition to the UK. It would have been useful to see how a similar process may be applied to these other countries to make this more generalizable. Of course, different data sources would be used for some of the inputs but there would be some similar data sources.-

It would be interesting to use the same model to model the cost effectiveness in Ireland and the Netherlands and compare the results to the UK; however, as our funding only covered an analysis of the UK data we were unable to include an analysis of the data from Ireland and the Netherlands in our study.

A separate study has been conducted for India; this has been carried out by researchers based at the Centre for Chronic Disease Control (CCDC) in India. The researchers developed a Markov Model to evaluate the impact of a polypill in an Indian population. A paper with the findings of the analysis is currently being drafted and will be submitted to a journal for peer review and publication (journal to be confirmed).

Methods

The inclusion criteria refer to those at high risk of CVD using the Framingham risk equation, however, later in the methods the QRisk2 is referred to as the preferred (better calibrated to the UK population, page 9) and it would be useful to reconcile why the different scores are used here.

We are using QRISK2 to define our model population and estimate CVD risk as our analysis is intended to inform decision making for the English NHS, and is funded on that basis. QRISK2 is currently recommended for use in the English population by the National Institute and Health and Care Excellence (NICE) and for the National Health Check policy. In practice the QRISK2 algorithm would be used to estimate an individual's risk of CVD and their requirement for CVD treatment. By using QRISK2 to define our population and estimate their baseline CVD risk we are aiming to reflect the population who would be likely to be prescribed the polypill in practice.

We are also planning to conduct a sensitivity analysis using the Framingham algorithm to estimate CVD risk for each individual who has not had a previous CVD event in the model and will compare the findings using Framingham compared to QRISK2.

It was not clear how adherence was measured within the trial or as the outcome in this model?

Perhaps this needs to be explained further.

The outcomes in the model are costs and quality of life measured used quality adjusted life years (QALYs). We will model via adherence to obtain these outcomes (see first sentence of section 3a in the paper). If the person is simulated to be adherent to medication, their risk of a CVD event will be lower as their risk of a CVD event will be modified by the treatment relative risk (Section 4). If a person has a CVD event in the model they will get an associated cost and reduction in quality of life. If they are adherent, they are likely to have less CVD events with an associated reduced cost and greater quality of life.

Adherence in the UMPIRE trial was defined as self-reported current use of antiplatelet, statin and ≥ 2 blood pressure lowering therapies for at least 4 days in the week preceding visits (baseline and end of trial visits). Systolic blood pressure (SBP) and LDL-cholesterol measurements were also taken at each visit. A sentence has been added to Page 3 (Line 18 and 19) defining how adherence was measured for the UMPIRE trial.

For modelling purposes, we are using Health Survey for England (HSE) data to estimate the probability of adherence in routine practice under usual care (i.e. without polypill). Participants in the HSE self-reported all the prescribed medications they had taken in the last 7 days (during interviews with the survey nurses). This data was coded in the HSE using the British National Formulary (BNF) classifications codes. Using this data, we were able to identify the medication patients were prescribed and whether or not they were taking the prescribed medication in the last week. A sentence on the data we will extract from the HSE in order to estimate individual adherence to medication in a usual care setting has been added to section 3a.

Section 3a outlines how the HSE data will be used to estimate the probability of each person being adherent or not to the indicated medications (antiplatelet, statins, antihypertensive). Section 3b details how we will estimate the relative risks of adherence to medication using UMPIRE trial data (UK data). The wording of Section 3b has been changed to make the second step clearer.

The probability in Section 3a will be multiplied by the relative risks of adherence estimated in Section 3b in the polypill scenario of the model. This will further modify their risk of a CVD event if they are adherent to medication. A sentence on this has been added to Section 3b.

The search strategy for models included a number of databases but was this sufficiently wide enough?

Before carrying out the literature review the search was discussed with a systematic reviewer based at the Health Economics Research Group (HERG) at Brunel University London. The systematic reviewer recommended the databases which were used. A larger search was not deemed necessary as the aim of the review was to identify the most appropriate modelling approach for our evaluation. It was felt that the databases chosen would pick up all relevant modelling approaches for our review. Further details on why we chose each database are below:

The NHS EED was identified as an appropriate database as this database reviews and produces critical commentaries economic evaluations of 'key' relevance to the UK NHS. The critical commentaries provide a summary of the overall reliability and generalisability of the study. The NICE HTA monograph series publishes research including cost-effectiveness analyses of healthcare treatment and tests; the series was searched to identify published HTA's which have developed or used a cardiovascular disease model. The NICE guidelines website was searched to identify guidelines related to cardiovascular disease (for example guidelines for lipid modification).

Further details on the review would have been helpful, even as a supplementary appendix. Also very little on the findings on the review were presented – instead the results suggest that the individual-level model simulation is the preferred method here.

A supplementary appendix has been added with further detail on the literature review. This includes a list of the identified models for review and their modelling approach (table 1), and an example of the data extraction form we used.

The aim of the review was to identify the best modelling approach for RUPEE. The review found that the two most popular modelling approaches are health state transition models (mainly Markov models) and simulation models. Each method was discussed in detail among the steering group for the study and the advantages and disadvantages of both approaches are discussed in the paper. Schematic illustrations of several models were prepared to aid discussions (example in supplementary appendix). It was decided that a simulation model was more appropriate for the RUPEE study as an individual simulation model can be used to capture the heterogeneity in the population allowing us to model individual CVD risk unlike a Markov model.

On page 8 adverse effects are referred to, but what about the different severity or consequences of these effects and combined effects of more than one adverse effect (or combined effect of multiple therapies on each AE)?

Adverse events in the model will include new onset diabetes from statins and antihypertensives (beta blockers and diuretics), cough from ace inhibitors and bleeding from aspirin. The severity for diabetes will be reflected in the costs (these will increase the longer a person has diabetes) and in the reduced quality of life. Cough and bleed will be more transient events with an associated cost and a reduction in quality of life. The associated costs and reduction in quality of life would capture the consequences of these adverse events.

The combined effect (effect on costs and reduction in quality of life) of 2 or more events happening at the same time will be estimated separately.

The costs for cough and bleed and reduction in quality of life are meant to reflect an average severity across the population.

Page 11, Line 21 and 22. The additional text 'including adverse events' has been added to the end of the sentence "Costs and QALYs will be recorded for each event" to reflect the consequences of these events in the main text.

Two different dosing strategies are mentioned for the polypill (version 1 and 2)– but the model described refers to one scenario of the polypill – how are the findings from the two versions to be combined? One version contains a diuretic instead of atenolol. These might expect to have different adverse effect profiles etc.

We are evaluating both versions of the polypill in the model, I have added "(polypill strategy will include polypill version 1 and version 2) to the end of the sentence "The RUPEE (NHS) model will be run twice, once to simulate costs and QALYs under usual care and once to simulate costs and QALYs under the polypill strategy" on Page 9.

How large is the HSE cross-sectional survey – further information on the representativeness of this for the model would be helpful. Was the adherence data self-reported from this?

We are using the Health Survey for England (HSE) 2011 dataset to get the population for our model. We used the 2011 database as the survey collected extra information on cardiovascular disease and diabetes in the population that year. We required the information on cardiovascular disease and diabetes for our model as this provided information on previous history of CVD events and associated medication. The HSE in 2011 provided a representative sample of the population at both national and regional level. A total of 8,610 adults and 2,007 children were interviewed. We will use the UMPIRE inclusion criteria to then select the model population from this population. Further detail on the HSE dataset can be found at <http://digital.nhs.uk/catalogue/PUB09300>

The adherence data from the HSE is self-reported. During the nurse visit for the HSE survey, participants are asked about all the prescribed medication they are currently taking (i.e. taking in the last 7 days). The nurses record this data using the British National Formulary (BNF) classification codes.

Further detail on the data we will extract from the HSE for the model including a summary table of the included population characteristics will be included in the second paper on this study which will report the findings of the economic model analysis.

There are reviews examining the predictors of adherence in CVD that could be referred to.

We have added references which we will use to help identify the individual characteristics which will be used as predictors of adherences. Further details on the individual characteristics we will use and their source(s) will be included in the second paper from this study which will report the findings of the RUPEE (NHS) study.

Section 4. 'the risk of a CVD event' will be adjusted by the relative risk of treatment, based on the treatment they are taking'. This was not clear – what treatment? Within the polypill? Other treatments patients may have received?

The treatment is medication for CVD – aspirin, statins and antihypertensives. The relative risk will be applied if the patient is taking and is adherent to that drug. We have changed the wording of the sentence in Section 4 to make this clearer.

Table 1 – For adverse events will the RR will be compared to 'no' treatment option in each case? Yes, for adverse events the relative risk will be compared with a no treatment option.

MI case fatality – based on data from one region in UK – is this sufficiently representative or has been used previously?

This is a good point, and we do agree that the data may lack national representativeness. However, we have not found any good UK national estimates of MI case fatality data and consider this source the best available data.

Costs of health states – some information given but not for all. More detail has been added to the costs section in Table 1.

Figure 1 – need to expand legend to Flowchart of literature review on CVD models perhaps. The legend has been expanded

Also, as mentioned above no information provided on what the review found.

A supplementary appendix has been included with some further detail.

Figure 2 has no legend and this needs to be included. Please provide full text for abbreviations used as footnotes etc.

The legend has been added and all abbreviations have been added as footnotes

Reviewer: 2

Page 4 lines 8-13: The guidelines mentioned were produced not by ISPOR acting as a body, but by a joint taskforce from ISPOR and the Society for Medical Decision Making. The description here should make that point clearly. Later (and in the abstract), it would be appropriate to refer to "the ISPOR-SMDM guidelines". This has been amended in the main text and abstract.

Page 4 line 46 should presumably be reference 12 not 11.- Thank you, this has been changed.

Page 7 lines 45-51: The reference to tunnel states is not really appropriate. Tunnel states deal with the specific issue of allowing the transition probabilities to vary according to time in a state. It would be better to make the general point that accounting for patient history requires multiplying the number of states to an infeasible level. This has been changed.

Page 8 line 15: PAD needs definition. This has been changed in the main text.

Page 11 line 9: the abbreviation AHT appears to be used only in Table 1 but never in the text. Accordingly, it should only be defined in the footnotes to the table, not here. The abbreviation has been removed from the main text

Table 1 footnote correctly repeats the definitions of some abbreviations given in the text, but not all of them. All abbreviations used in the table should be defined in its footnote. All abbreviations have been added to the footnotes for Table 1

VERSION 2 – REVIEW

REVIEWER	Kathleen Bennett
REVIEW RETURNED	24-Aug-2016

GENERAL COMMENTS	No further comments
---------------------

REVIEWER	Pelham Barton
REVIEW RETURNED	05-Sep-2016

GENERAL COMMENTS	I have checked the responses to my previous review and am happy with the way these have been made.
--